# A Novel Intragenic Duplication in the *HDAC8* Gene Underlying a Case of Cornelia de Lange Syndrome

**DOI:** 10.3390/genes13081413

**Published:** 2022-08-08

**Authors:** Cristina Lucia-Campos, Irene Valenzuela, Ana Latorre-Pellicer, David Ros-Pardo, Marta Gil-Salvador, María Arnedo, Beatriz Puisac, Neus Castells, Alberto Plaja, Anna Tenes, Ivon Cuscó, Laura Trujillano, Feliciano J. Ramos, Eduardo F. Tizzano, Paulino Gómez-Puertas, Juan Pié

**Affiliations:** 1Unit of Clinical Genetics and Functional Genomics, Department of Pharmacology-Physiology, School of Medicine, Universidad de Zaragoza, CIBERER-GCV02 and IIS-Aragon, E-50009 Zaragoza, Spain; 2Department of Clinical and Molecular Genetics Hospital Vall d’Hebron, E-08035 Barcelona, Spain; 3Medicine Genetics Group, Vall Hebron Research Institute, E-08035 Barcelona, Spain; 4Centro de Biología Molecular Severo Ochoa, CBMSO (CSIC-UAM), Molecular Modeling Group, E-28049 Madrid, Spain; 5Department of Genetics, Hospital Sant Pau, E-08041 Barcelona, Spain; 6Unit of Clinical Genetics, Department of Paediatrics, Service of Paediatrics, Hospital Clínico Universitario Lozano Blesa, School of Medicine, Universidad de Zaragoza, CIBERER-GCV02 and IIS-Aragon, E-50009 Zaragoza, Spain

**Keywords:** Cornelia de Lange syndrome, genetic disorder, copy number variants, *HDAC8*, intragenic duplication, array CGH, genetic diagnosis

## Abstract

Cornelia de Lange syndrome (CdLS) is a multisystemic genetic disorder characterized by distinctive facial features, growth retardation, and intellectual disability, as well as various systemic conditions. It is caused by genetic variants in genes related to the cohesin complex. Single-nucleotide variations are the best-known genetic cause of CdLS; however, copy number variants (CNVs) clearly underlie a substantial proportion of cases of the syndrome. The *NIPBL* gene was thought to be the locus within which clinically relevant CNVs contributed to CdLS. However, in the last few years, pathogenic CNVs have been identified in other genes such as *HDAC8*, *RAD21*, and *SMC1A*. Here, we studied an affected girl presenting with a classic CdLS phenotype heterozygous for a de novo ~32 kbp intragenic duplication affecting exon 10 of *HDAC8*. Molecular analyses revealed an alteration in the physiological splicing that included a 96 bp insertion between exons 9 and 10 of the main transcript of *HDAC8*. The aberrant transcript was predicted to generate a truncated protein whose accessibility to the active center was restricted, showing reduced ease of substrate entry into the mutated enzyme. Lastly, we conclude that the duplication is responsible for the patient’s phenotype, highlighting the contribution of CNVs as a molecular cause underlying CdLS.

## 1. Introduction

Cornelia de Lange syndrome (CdLS, OMIM #122470, #300590, #610759, #614701, #300882) is a genetically heterogeneous multisystemic disorder with an estimated prevalence of one in 10,000 to 30,000 live births. Since severity and clinical manifestations can vary widely among individuals, the CdLS phenotype has been defined as a spectrum that includes the so-called “classic” and “nonclassic” clinical presentations. The most clinically recognizable findings include distinctive facies with synophrys (HP:0000664), concave nasal bridge (HP:0011120), upturned nasal tip (HP:0000463), smooth philtrum (HP:0000319), and thin upper lip vermilion (HP:0000219), all of which are helpful in the diagnostic approach. Patients also commonly show intellectual disability (HP:0001249), prenatal and postnatal growth retardation (HP:0001511, HP:0008897), microcephaly (HP:0000252), limb reduction defects (HP:0001180, HP:0009237), and hirsutism (HP:0001007) [1].

From a molecular point of view, CdLS has been linked to eight genes involved in the structural or regulatory function of the cohesin complex. The most frequent causal gene is *NIPBL*, followed by *SMC1A* and *HDAC8*, and less frequently *SMC3*, *RAD21*, *BRD4*, *MAU2*, and *ANKRD11* [1]. Although a consistent link between the severity of the phenotype and the type of the genetic change has not been revealed so far, some clinical features have been associated more frequently with genetic variants in specific genes [2,3,4,5]. For example, patients with pathogenic variants in *HDAC8* have facial and clinical distinguishable features such as ocular hypertelorism (HP:0000316), delayed fontanelle closure (HP: 0000270), bulbous nasal tip (HP:0000414), hooding or redundant overfolded skin of the upper eyelids, dental anomalies, and mosaic skin pigmentation [6,7].

Currently, the widespread use of sequencing targeted panels, including causative and related CdLS genes, has significantly improved the diagnosis success rate, as well as reducing the time to achieve it [8]. However, although the causal variant of a CdLS case involves only one of the related genes, genetic diagnosis may still be challenging due to difficulties in interpretation such as allele frequency or even mosaicism, which appear to be quite recurrent in CdLS [9]. Furthermore, the genetic variant type can range from single-nucleotide variants (SNVs) to small insertions and deletions (INDELs) or copy number variants (CNVs). In fact, the presence of pathogenic CNVs in *NIPBL* may account for up to 3% of CdLS cases [10]. Therefore, the international guidelines recommend multiplex ligation-dependent probe amplification (MLPA) approaches when panels and Sanger sequencing cannot detect any variant in this gene [1], but commercial MLPA assays only cover the *NIPBL* gene. Since next-generation sequencing panels have been implemented, structural variants involving other causal genes such as *SMC1A* [11], *HDAC8* [12], and *RAD21* [13] have been described in some individuals with CdLS. These cases indicate that pathogenic CNVs in CdLS-related genes may be more common than previously thought.

In this study, we report for the first time a molecular functional study of an intragenic duplication in the *HDAC8* gene identified in a girl with classic CdLS phenotype. We present genotype data and assess the pathogenicity of the intragenic variant through a combination of clinical phenotype evaluation, array CGH with exonic coverage of several genes involved with CdLS, splicing analysis, and structural prediction with protein modeling.

## 2. Materials and Methods

### 2.1. Clinical Diagnosis

The study was performed according to the Declaration of Helsinki protocols and was approved by each Regional Ethics Committee of Clinical Research. Informed consent was obtained from parents or guardians of all individuals included in this study. Additional informed consent was collected for the publication of photographs of the patient. Clinical data were collected by a clinical geneticist at the Vall d’Hebron Hospital (Barcelona) following a standard restricted-term questionnaire. The clinical score was calculated by CdLS clinical geneticist specialists according to the international consensus guidelines [1]. Face2Gene (https://www.face2gene.com (accessed on 10 January 2022)) was used to determine the most probable clinical diagnoses for the patient [14].

### 2.2. Isolation of DNA and RNA

Sequencing analyses were carried out on DNA from the patient and her progenitors. The DNA was isolated from non-cultivated blood samples using a Gentra^®^ Puregene^®^ Kit (Qiagen, Hilden, Germany) following the recommendations of the manufacturer.

Total RNA was extracted from 10 mL of peripheral blood lymphocytes using Trisure reagent (Sigma-Aldrich, St. Louis, MO, USA) following the manufacturer’s protocol. RNA was cleaned up using an RNeasy Mini Kit (Qiagen) with an additional step of DNase digestion using an RNase-Free DNase Set (Qiagen). Purity and integrity of the RNA were assessed by electrophoresis and spectrophotometry using a NanoDrop ND-2000 spectrophotometer (NanoDrop Technologies, Wilmington, DE, USA).

### 2.3. Next-Generation Sequencing

The patient´s DNA was analyzed on a panel of gene amplicons specifically designed for CdLS in the Clinical Genetics and Functional Genomics Group at the University of Zaragoza, as previously described [9]. The variants were classified according to the ACMG recommendations and detailed information provided in the public databases gnomAD (https://gnomad.broadinstitute.org/ (accessed on 7 March 2022)), ClinVar (https://www.ncbi.nlm.nih.gov/clinvar/ (accessed on 7 March 2022)), dbSNP (https://www.ncbi.nlm.nih.gov/snp/ (accessed on 7 March 2022)), LOVD (https://www.lovd.nl/ (accessed on 7 March 2022)), and relevant scientific literature. The in silico analyses were performed using the following online tools: Polyphen-2 (http://genetics.bwh.harvard.edu/pph2/ (accessed on 7 March 2022)), SIFT (https://sif.bii.a-star.edu.sg/ (accessed on 7 March 2022)), and the integration tool VarSome (https://varsome.com/ (accessed on 7 March 2022).

### 2.4. Array Comparative Genomic Hybridization

DNA extracted from an uncultured blood sample was analyzed with CytoSure Constitutional 8× 60K v3 (Oxfordshire, UK) array comparative genomic hybridization (array CGH) following the recommendations of the manufacturer. CytoSure Constitutional 8× 60K v3 has exonic resolution in 354 genes selected by the ClinGen Dosage Sensitivity Map192, including *SMC1A*, *HDAC8*, *RAD21*, and *ANKRD11*. CNVs were classified following recommendations of the American College of Medical Genetics and Genomics standards [15] and reevaluated with actualized guidelines [16].

### 2.5. cDNA Synthesis and Analysis

RNA isolated from blood lymphocytes was reverse-transcribed using random hexamer primers with an Invitrogen SuperScript™ First-Strand Synthesis System Kit for qPCR. For the analysis of physiological splicing of *HDAC8* (ENST00000373573.9, NM_018486.3), the cDNA was amplified with different pairs of primers using a Thermo Scientific™ DreamTaq PCR Master Mix (2×) Kit in Applied Biosystems equipment. Primers were designed using the Primer3Plus in silico tool (https://www.bioinformatics.nl/cgi-bin/primer3plus/primer3plus.cgi (accessed on 7 March 2022 )) and checked using the UCSC in silico PCR tool (https://genome.ucsc.edu/cgi-bin/hgPcr (accessed on 7 March 2022)). The reverse primer spanned the junction between exons 10 and 11, and it was the same for all PCRs with the sequence 5´GCTTCAGATTCCCTTTGATGTAG 3´ (Reverse 1). The forward primer was different for each PCR: Forward 1 bound within exons 1 and 2, 5´CAAACGGGCCAGTATGGTG 3´; Forward 2 bound within exons 7 and 8, 5´GATTTTTCCCAGGAACAGGT 3´; Forward 3 hybridized with exon 9, 5´GAGGCTATAACCTTGCCAAC 3´.

PCR products were purified by NZYtech NZYGelpure Kit and screened by Sanger sequencing on ABI3730xl Capillary Electrophoresis Sequencing System according to the manufacturer´s protocol.

### 2.6. Real-Time Quantitative PCR (qPCR)

Relative quantification of *HDAC8* expression was performed by qPCR. In this experiment, we used six cDNA samples as control to compare with *HDAC8* expression. After cDNA synthesis, qPCR amplification was carried out using the Applied Biosystems SYBR™ Green PCR Master Mix Kit on an Applied Biosystems™ QuantStudio™ 5 System. We used the following amplification conditions: 95 °C for 10 min for the hold stage; 40 cycles at 95 °C for 15 s, 60 °C for 1 min; finally, 95 °C for 15 s, 60 °C for 1 min, and 95 °C for 20 s for the melt curve stage. Samples were assessed in triplicate. 

The primer sequences were as follows: Forward 8–9 5´TTGGGAGGAGGAGGCTATAAC 3´ and Reverse 9–10 5´GCTGTGAAAAACTCATGATCTGG 3´; Forward 9-Insertion 5´CCAGATCATGAGAATATGCCTG 3´ and Reverse Insertion-10 5´CTGTGAAAAACTGCACATCACA 3´; Forward Exon 1 5´ CGCTGGTCCCGGTTTATATC 3´ and Reverse Exon 2 5´ TGCAGTGCATATGCTTCAATC 3´. Gene expression levels were calculated normalizing with respect to the housekeeping gene β-actin, using the forward β-actin 5´ CTTCCTGGGCATGGAGTC 3´ and reverse β-actin 5´ AGCACTGTGTTGGCGTACAG 3´ primers. 

The Ct values for each sample were determined with amplification plots in the logarithmic phase. The PCR outcome and efficiency of amplification were determined using QuantStudio™ Design and Analysis Software (v1.5.1, Applied Biosystems, Waltham, MA, USA) using the 2^−∆∆C^t method. GraphPad Prism was used for the graphics.

### 2.7. Structure Modeling of HDAC8 Variant and Molecular Dynamics Simulation

The 3D structure of the *HDAC8* variant protein was modeled using the crystal structure of human *HDAC8* (PDB ID: 1T64 [17]) as a template. The model was built using the SWISS-MODEL server (http://swissmodel.expasy.org (accessed on 5 April 2022)), its structural quality being within the range accepted for homology-based structures (Anolea/Gromos/QMEAN4). Structures for wildtype and variant *HDAC8* proteins were subjected to 200 ns of unrestrained molecular dynamics (MD) simulation using the AMBER18 molecular dynamics package (http://ambermd.org/ (accessed on 5 April 2022); University of California—San Francisco, CA, USA), essentially as previously described [18]. In brief, 3D models were first solvated with a periodic octahedral pre-equilibrated solvent box using the LEaP module of AMBER, with 12 Å as the shortest distance between any atom in the protein domain and the periodic box boundaries. Free MD simulation was performed using the PMEMD program of AMBER18 and the ff14SB force field (http://ambermd.org/ (accessed on 5 April 2022)), applying the SHAKE algorithm, a time step of 2 fs, and a nonbonded cutoff of 12 Å. Systems were initially relaxed over 10,000 steps of energy minimization, using 1000 steps of steepest descent minimization followed by 9000 steps of conjugate-gradient minimization. Simulations were then started with a 20 ps heating phase, raising the temperature from 0 to 300 K in 10 temperature change steps, after each of which velocities were reassigned. During minimization and heating, the Cα trace dihedrals were restrained with a force constant of 500 kcal·mol^−1^·rad^−2^ and gradually released into an equilibration phase in which the force constant was progressively reduced to 0 over 200 ps. After the equilibration phase, 200 ns of unrestricted MD simulation was obtained for the structures. MD trajectories were analyzed using VMD software (v1.9.3., University of Illinois, Urbana, USA) [19]. Figures were generated using the Pymol Molecular Graphics System (https://pymol.org/ (accessed on 5 April 2022); Schrödinger, LLC, Portland, OR, USA).

## 3. Results

### 3.1. Clinical Report

The proband is a 5 year old girl, the first child of nonconsanguineous healthy parents. She has a healthy younger sibling. During pregnancy, intrauterine growth restriction (IUGR) was revealed in the second trimester. The patient was born at 41 weeks gestational age via spontaneous vaginal delivery (SVD). At that moment, it was noted that she showed congenital microcephaly (HP:0000252) (head circumference 32.0 cm; −2.07 SD) and symmetrical IUGR (HP:0001511) with a birthweight of 2.230 kg (−2.79 SD) and length of 45 cm (−3.22 SD). Growth retardation (HP:0008897) and microcephaly (HP: 0000252) persisted after birth, and, at the age of 22 months, head circumference was 43.5 cm (−3.62 SD), weight was 9.5 kg (−1.92 SD), and length was 76.5 cm (−2.81 SD). General motor development was not significantly delayed, and she achieved sitting and walking independently at 9 and 16 months, respectively. However, at the present age of 5 years, she is currently still nonverbal and shows global developmental delay (HP:0001263) and behavioral problems. Regarding dysmorphic facial features, she showed bulbous nasal tip (HP:0000414), long philtrum (HP:0000319), synophrys (HP:0000664), microdontia (HP:0000691), and widely spaced maxillary central incisors (HP:0001566). Furthermore, she presented delayed closure of fontanels (more than 2 years) (HP: 0000270), hirsutism (HP:001007), small hands (HP:0200055) and feet (HP:0001773), short fifth finger (HP:0009237), clinodactyly of the fifth finger (HP:0004209), sensorineural hearing impairment (HP:0000407), and gastroesophageal reflux (HP:0002020) (Figure 1A). On the basis of these features, an expert clinical geneticist assigned the clinical diagnosis of CdLS during early childhood with a clinical score of 12. An additional clinical analysis was carried out with Face2Gene^®^ at the age of 5, and KBGS and CdLS were the first and second syndromes suggested, respectively, with a medium–high probability.

### 3.2. DNA Molecular Analyses

An initial genetic test with a CdLS deep targeted gene panel (>1000×) did not detect any potentially constitutive and/or mosaic causative genetic variant in DNA from blood. Somatic mosaic variants could not be ruled out totally since it was not possible to obtain another biological sample from the patient. A variant of uncertain significance was identified in the ANKRD11 gene ((NM_001256183.1; c.890C > T, p.(Thr297Met)) (G = 1019, A = 976). However, although some in silico predictors such as SIFT (0.0) and Polyphen-2 (0.999) suggested a possible damaging effect, this variant is reported as likely benign in ClinVar, and the allele frequency in gnomAD is greater than 0.001. Furthermore, familiar co-segregation studies revealed that the healthy mother carried the variant. Therefore, despite the compatible genotype–phenotype correlation in the patient, this variant was reclassified as likely benign according to ACMG criteria.

Despite the compatible genotype–phenotype correlation, familiar segregation studies revealed that the healthy mother carried the variant; therefore, the causality of this one was ruled out. Oligonucleotide-based array CGH was subsequently performed on genomic DNA using CytoSure Constitutional v3 array 8 × 60 K, which offers enhanced exon-level coverage of 354 developmental disorder genes. A duplication spanning ~32 kbp at Xq13.1 was identified in the patient (arr(GRCh38) Xq13.1(72,340,096–72,389,392) × 3) (Figure 1B and Figure 2A). This duplication implicates at least exon 10 (NM_018486.3) of the *HDAC8* gene and was not present in leucocyte-derived DNA of the parents.

### 3.3. RNA Molecular Analyses

To assess the functional impact of the intragenic duplication on *HDAC8* transcription, we performed qPCR and Sanger sequencing from blood cDNA. Specific quantitative amplification of the pair spanning exons 1–2 and 9–10 revealed a significant reduction in exon junction 9–10 in the patient (Figure 2B). We also performed a conventional PCR amplification from exon 9 to exon junction 10–11. An aberrant transcript of 309 bp in addition to the expected wildtype PCR product of 213 bp was observed. Sequencing analysis of the aberrantly spliced product revealed an insertion of a 96 bp fragment between exons 9 and 10 (Figure 2C and Appendix A). This sequence aligns with an intronic region located between 225,627 bp and 225,722 bp of intron 9 of *HDAC8*, GRCh38.p13 chrX: 72,352,477–72,352,382. In addition, we confirmed the presence of the 96 bp insertion in the patient by qPCR (Figure 2D).

### 3.4. Structural Prediction of HDAC8 Variant

A theoretical 3D structure for the *HDAC8* variant was obtained by homology modeling. As a result of the 96 bp insertion in the nucleotide sequence, at the structural level, amino acids 336 to 377 of the C-terminus of the protein were replaced, including a complete α helix (in red in Figure 3A, left) by a much shorter segment of six amino acids (in red in Figure 3A, right), followed by a stop codon. To analyze the effect that this C-terminal deletion could have on the structure and function of the enzyme, the wildtype and variant protein models were simulated for 200 ns of unrestricted molecular dynamics, comparing the behavior of the two structures. The main differences between them were, on the one hand, a marked change in the surface electrostatic charge around the entrance of the active center, from being mostly electropositive in the wildtype protein (blue-colored patch in Figure 3B, left) to neutral in the variant protein (light-blue and white colors in the equivalent position in Figure 3B, right). In addition to this shift in surface charge, a notable difference observed between the two was a displacement in the variant protein, starting at approximately 140 ns, of the loop containing amino acids Gly206 to Gly220, located at the entrance of the active center, shortening the distance between this loop and the opposite wall of the substrate entry site (distance marked with an arrow in Figure 3A, left and right). As a quantification of this movement, Figure 3C shows the variation in the distance between the α carbon of the Pro209 and Gly151 residues, located in the shifting loop and in a loop on the opposite wall of the entry site, respectively, decreasing from an average of 8.5 Å in the wildtype protein to 5.5 Å in the variant *HDAC8* protein.

## 4. Discussion

Identification of the genetic cause sometimes remains a challenge in CdLS. Although this is a phenotypically recognizable syndrome, the huge clinical and genetic heterogeneity makes it difficult to establish genotype–phenotype correlations. Here, we described the genetic diagnosis procedure of a girl with classic CdLS phenotype and highlighted the limitations faced during this process: (i) interpretation of VUS, (ii) technical difficulty in detecting small CNVs and somatic mosaic variants, and (iii) functional interpretation of intragenic duplications. In this study, we presented a novel intragenic ~ 32 kbp duplication affecting exon 10 of *HDAC8* gene. CdLS-associated variants in the *HDAC8* gene (Xq13.1) are estimated to account for the 4% of the cases of this disorder (CDLS5, OMIM #300882).

The majority of known disease-causing changes in *HDAC8* are SNVs, including nonsense, missense, or splice site variants [6,7,20,21,22,23,24,25]. Nevertheless, recently, several cases of changes involving larger regions of *HDAC8* have been reported in individuals with CdLS, especially intragenic deletions ranging from single to multiple exons [6,12]. Interestingly, the presence of two to three pairs of microhomology at the breakpoints was found in these cases [12]. To date, only one intragenic duplication in *HDAC8* has been reported in the literature (GRCh38.p13 (chrX: 72,371,425–72,731,334) × 3), whose region affected includes exons 6 to 9 (p.Phe336Leufs*1) (6). In addition, ClinVar reports one likely pathogenic duplication (ClinVar: VCV000442981.2, GRCh38.p13 (chrX: 72,348,432–72,549,544) × 3), and DECIPHER reports two duplications associated with patients with overlapping CdLS phenotypes (Patient 275,487, Patient 275,258) (Figure 2A). However, to our knowledge, no report of the effect of an intragenic duplication in *HDAC8* at a molecular level has been provided.

Haploinsufficiency for genes within a deletion CNV is a well-recognized cause of genetic disease. However, duplication CNVs might cause disease through triplosensitivity, gene disruption, or gene fusion at breakpoints. Undoubtedly, interpreting the genetic consequences of the duplication is essential to understand the etiology of the genetic disease. In this report, we presented the case of an intragenic duplication in *HDAC8* that disrupted the reading frame of at least the most common *HDAC8* gene isoform present in blood. By conventional splicing analysis and qPCR, we detected an aberrant *HDAC8* transcript in the patient. Surprisingly, this transcript included an insertion between exons 9 and 10, corresponding to a 96 bp fragment of intron 9. By analyzing this intronic sequence, we found that it contained noncanonical splice site sequences, AG donor (5′) and GC acceptor (3′) [26]. As a consequence, the translation was impaired, resulting in a protein of 342 amino acids instead of the 377 comprising the wildtype protein (Appendix A) due to a break in the reading frame caused by the insertion of a premature stop codon.

Although the direct correlation of clinical severity with the activity of *HDAC8* variants is complex, the structure–function relationship has been previously proven. Depending on the structural location of the different variants studied in this enzyme, the degree of loss of enzyme activity is variable, being greater when the catalytic center is affected or when the mutations affect an amino acid located in a conserved region of the protein [27,28,29,30]. All the mutations previously analyzed in *HDAC8* were randomly distributed throughout the entire protein structure, and, in the vast majority of them, the enzyme activity was compromised [6,27,28,29,30,31]. The de novo intragenic duplication presented in the current studio involved only exon 10, which, at the structural level, mainly affected an α helix at the C-terminal end of the protein, far from the active center. 

With such a large deletion within the carboxyl-terminal end, a drastic effect was expected in the molecular dynamics simulation, entailing a large disorganization of the protein. However, it seems striking that the overall structure of the wildtype and variant proteins remained almost constant throughout 200 ns of simulation (Figure 3A). Despite this, notable differences were observed between them, such as a change in the surface charge next to the entry of the active center in the variant protein (Figure 3B), which would compromise the correct binding between the mutated enzyme and the substrate. This alteration would also trigger a displacement of a loop near the entrance of the catalytic site, which remains stable at the new position (Figure 3C) and results in reduced accessibility of the substrate to the active center (Figure 3B). Experimental *HDAC8* activity measurements would be necessary to verify this hypothesis.

In summary, the progressive increase in reports of patients with CdLS and intragenic CNVs makes it necessary to cover a wider genetic diagnosis scenario. For this purpose, as NGS becomes routine for genetic testing, we must incorporate specific designs and pipelines that analyze and detect this type of variant, not only in the *NIPBL* gene but also in the other causal genes of CdLS such as *HDAC8*. Lastly, a comprehensive molecular characterization of the intragenic duplications is essential for the investigation of their pathogenicity, with the ultimate goal of providing a better understanding of the disease.

## Figures and Tables

**Figure 1 genes-13-01413-f001:**
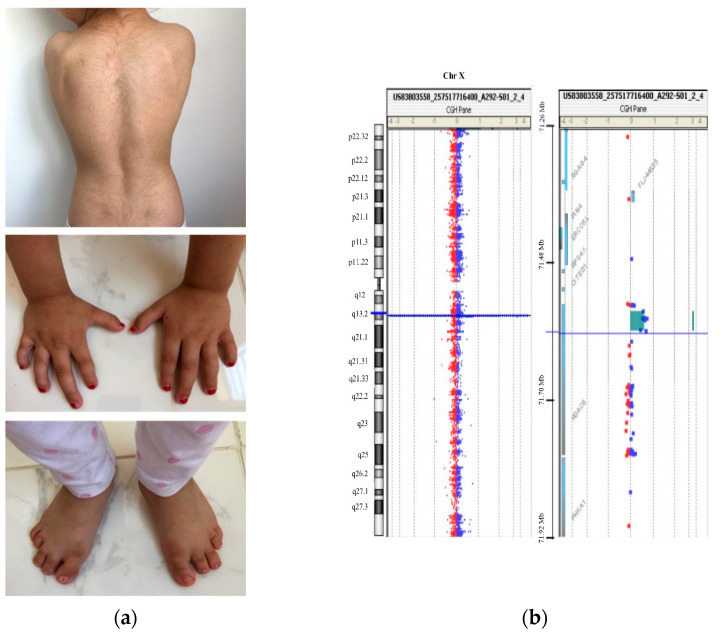
Clinical and genetic description of the patient. (**a**) Patient at 5 years showing her back with hirsutism and small hands and feet with clinodactyly of the fifth finger (more details in the text); (**b**) CytoSure Constitutional v3 array 8 × 60K array-CGH result showing a 0.032 Mb duplication at Xq13.1 in the patient.

**Figure 2 genes-13-01413-f002:**
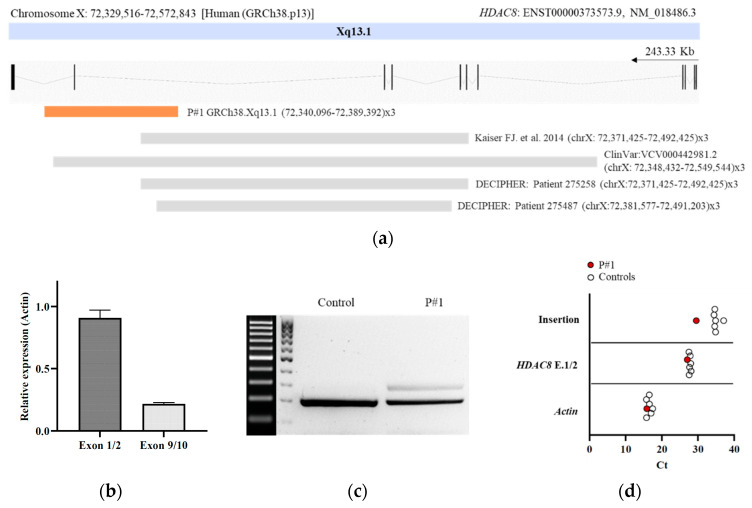
Molecular characterization of the intragenic duplication in *HDAC8*. (**a**) Schematic representation of intragenic duplications on *HDAC8* related to CdLS reported to date. The genomic region duplicated for each case is shown. Localizations of duplications on *HDAC8* are indicated by chromosome band and position (human (GRCh38.p13)). The arrow indicates the direction of transcription. The region duplicated in the patient is marked by an orange-filled box [6]; (**b**) qPCR analysis of exons 1–2 and 9–10 of *HDAC8* (NM_018486.3). Gene expression levels were normalized to actin. The expression level in controls was arbitrarily set to 1.0; (**c**) agarose gel of the cDNA PCR products. After amplification of a fragment compromising exons 9 and 10, cDNA of the patient yielded the expected PCR product of 213 bp, as well as an aberrant fragment of 309 bp corresponding to an insertion of 96 bp between exons 9 and 10; (**d**) Cts from qPCR analysis of exons 1–2 and the Δ96 bp region in the patient and controls. Gene expression levels were normalized to *Actin*.

**Figure 3 genes-13-01413-f003:**
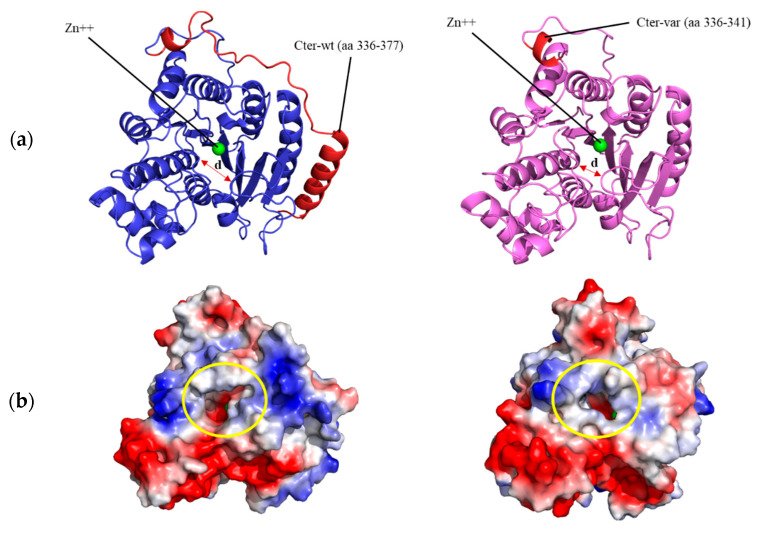
Structural modeling of *HDAC8* variant. (**a**) Structural model of wildtype (**left**) and variant (**right**) human *HDAC8* proteins after 200 ns of unrestricted molecular dynamics simulation. The differential C-terminal end between the two proteins is colored in red. The position of the Zn^++^ atom in the active center and the distance between residues Pro209 and Gly151 (arrow) are indicated; (**b**) surface of both proteins after 200 ns of molecular dynamics. The entrance of the active center is circled in yellow. The surface is colored according to the electrostatic charge (red: negative, blue: positive); (**c**) distance in Å between the α carbon atoms of residues Pro209 and Gly151 over the 200 ns simulation of the wildtype (wt) and variant (var) *HDAC8* protein.

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
