# Peer review of "A Novel Intragenic Duplication in the HDAC8 Gene Underlying a Case of Cornelia de Lange Syndrome"

_genes, 2022, doi:10.3390/genes13081413_

Round 1

Reviewer 1 Report

The authors present their case of an affected girl presenting with a classic CdLS phenotype who is heterozygous for a de novo Ì´ 32 Kb intragenic duplication affecting exon 10 of HDAC8. In addition, they report several functional analyses underling this duplication as causative for the CdLS phenotype.

This is a very well written manuscript with a straightforward methodology, clear figures and a correct interpretation of the literature. Therefore, I believe this manuscript can be accepted after making some slight modifications/ additional analyses+elaboration (see below).

Some suggestions: 

-        in the introduction “such as allele frequency or even mosaicism, that appears quite recurrent in CdLS ” => is there a way to overcome this issue of mosaicism? NGS? type of cells? 

-        is it by a decrease of protein levels or decrease of enzyme activity

-        3.1: “younger sibling” => is this sibling tested? the risk to the sibs of a proband with CdLS is estimated to be 1.5% due to the possibility of germline mosaicism…

-        “during early childhood with a clinical score of 12” is this scoring based on “Kline AD, Moss JF, Selicorni A, Bisgaard AM, Deardorff MA, Gillett PM, Ishman SL, Kerr LM, Levin AV, Mulder PA, Ramos FJ, Wierzba J, Ajmone PF, Axtell D, Blagowidow N, Cereda A, Costantino A, Cormier-Daire V, FitzPatrick D, Grados M, Groves L, Guthrie W, Huisman S, Kaiser FJ, Koekkoek G, Levis M, Mariani M, McCleery JP, Menke LA, Metrena A, O'Connor J, Oliver C, Pie J, Piening S, Potter CJ, Quaglio AL, Redeker E, Richman D, Rigamonti C, Shi A, Tümer Z, Van Balkom IDC, Hennekam RC. Diagnosis and management of Cornelia de Lange syndrome: first international consensus statement. Nat Rev Genet. 2018 Oct;19(10):649-666. doi: 10.1038/s41576-018-0031-0. PMID: 29995837; PMCID: PMC7136165.” => if yes, please include this reference; if not, please elaborate on the methog of scoring

-        regarding the VUS in ANKRD11: “that the healthy mother carried the variant, and therefore the causality of this one was ruled out.”=> There is significant variability in the clinical findings, even between affected members of the same family. Hence, segregation considering only the mother does not seem to be sufficient. I would recommend further segregation at the maternal side of the family. Moreover, I would recommend in silico prediction tools to potentially classify this variant to a class 2 (likely benign), e.g. possible by using http://wintervar.wglab.org/ ; how about clinvar, gnomAD,…?

-        As a consequence, the translation is impaired, resulting in a protein of 342amino acids instead of the 377 composing the wild-type protein (Table S2 and Table S3) due to a break in the reading frame caused by the insertion of a premature stop codon.” => very interesting finding! 

-        especially intragenic deletions ranging from single to multiple exons [6,12]. Interestingly, the presence of two to three pairs of microhomology at the breakpoints was found in these cases [12].” => it would be interesting to make a summary/tableof all the patients with CdLS and intragenic duplication to see which genes are affected by which duplication and how this affects the phenotype?

-        discussion: any limitation(s) of the study? how about the VUS in ANKRD11?... please briefly elaborate.

Author Response

Dear Editor,

We sincerely thank to the reviewers for the careful and detailed revision of our manuscript. We have submitted a revised version of the manuscript, addressing the concerns/doubts raised by the referees. Any changes made to the original manuscript have been incorporated using the "Track Changes" function in Microsoft Word. We strongly believe that our manuscript has been improved after the current revision. Please, find below the point-by-point responses to the comments and concerns raised by the reviewers.

We hope these changes will be to your liking.

Best regards,

Prof. Juan Pié, PhD        

Response to Reviewer 1 Comments

The authors present their case of an affected girl presenting with a classic CdLS phenotype who is heterozygous for a de novo Ì´ 32 Kb intragenic duplication affecting exon 10 of HDAC8. In addition, they report several functional analyses underling this duplication as causative for the CdLS phenotype.

This is a very well written manuscript with a straightforward methodology, clear figures and a correct interpretation of the literature. Therefore, I believe this manuscript can be accepted after making some slight modifications/ additional analyses+elaboration (see below).

We sincerely thank to the reviewer for the careful and thorough revision of our manuscript. We thank you very much for the interesting comments and suggestions.

We have tried to take into account all the suggested changes as well as show the limitations of the study. 

We will kindly answer all suggestions point by point below. We truly appreciate your revision; these changes improve our manuscript.

Point 1:

In the introduction “such as allele frequency or even mosaicism, that appears quite recurrent in CdLS ” => is there a way to overcome this issue of mosaicism? NGS? type of cells?

Response 1:

The high prevalence of mosaicism in CdLS should be considered when molecular diagnosis of the proband and familiar co-segregation studies are planned. An invaluable tool to reach this high percentage of solved cases is sensitive next generation sequencing, and in particular the incorporation of deep-sequencing target panels. Furthermore, the tissue analyzed is important, not only do we have to analyze the standard routine sample such as blood, but we also have to include other samples such as buccal cells, saliva, urine, fibroblasts and/or skeletal muscle. This issue has been studied and analyzed in depth in a previous work of our group (Latorre-Pellicer A. et al., Sci. Rep. 2021, 11, 15459).

In this study we have perform a deep sequencing panel that reach more than 1000 x in order to detect a possible genetic mosaicism. However, only blood samples were available, and we were unable to analyze other biological sample. We have included this limitation in line 221:

Line 226: An initial genetic test with a CdLS deep targeted gene panel (> 1000x) did not detect any potentially constitutive and/or mosaic causative genetic variant in DNA from blood. Somatic mosaic variants could not be rule out totally since it was not possible to obtain other biological sample from the patient. A variant of uncertain significance was identi-fied in the ANKRD11 gene [(NM_001256183.1; c.890C>T, p.(Thr297Met)] (G=1019, A=976).

Point 2:

Is it by a decrease of protein levels or decrease of enzyme activity

Response 2:

In silico analyses suggest that there is a decrease in enzyme activity due to the conformational change of the variant protein. As a consequence, the entrance of the active center is altered and as a result the activity is affected (Results: lines 257- 278; Discussion: lines 341 to 350). However, we have not been able to experimentally verify this point due to the lack of a sample to carry out the study. So, we have included this limitation in the discussion:

Line 374: Experimental HDAC8 activity measurements would be necessary to verify this hypothesis.

Point 3:

3.1: “younger sibling” => is this sibling tested? the risk to the sibs of a proband with CdLS is estimated to be 1.5% due to the possibility of germline mosaicism…

Response 3:

We accomplished the genetic study only on the patient and her parents. The sibling did not present any features suggestive of the syndrome, and although the possibility of germinal mosaicism must be taken into account, family has no plans to have more children.

Point 4:

“during early childhood with a clinical score of 12” is this scoring based on “Kline AD, Moss JF, Selicorni A, Bisgaard AM, Deardorff MA, Gillett PM, Ishman SL, Kerr LM, Levin AV, Mulder PA, Ramos FJ, Wierzba J, Ajmone PF, Axtell D, Blagowidow N, Cereda A, Costantino A, Cormier-Daire V, FitzPatrick D, Grados M, Groves L, Guthrie W, Huisman S, Kaiser FJ, Koekkoek G, Levis M, Mariani M, McCleery JP, Menke LA, Metrena A, O'Connor J, Oliver C, Pie J, Piening S, Potter CJ, Quaglio AL, Redeker E, Richman D, Rigamonti C, Shi A, Tümer Z, Van Balkom IDC, Hennekam RC. Diagnosis and management of Cornelia de Lange syndrome: first international consensus statement. Nat Rev Genet. 2018 Oct;19(10):649-666. doi: 10.1038/s41576-018-0031-0. PMID: 29995837; PMCID: PMC7136165.” => if yes, please include this reference; if not, please elaborate on the methog of scoring

Response 4:

In point 2.1. Clinical diagnosis we indicate how we calculated the clinical score and we refer to the reference kindly provided by the reviewer.

Line 92: The clinical score was calculated by clinical geneticists specialized in CdLS according to the international consensus guidelines [1].

[1] Kline, A.D.; Moss, J.F.; Selicorni, A.; Bisgaard, A.M.; Deardorff, M.A; Gillett, P.M.; et al. Diagnosis and management of Cornelia de Lange syndrome: first international consensus statement. Nat. Rev. Genet. 2018, 19, 649-666.

Point 5:

Regarding the VUS in ANKRD11: “that the healthy mother carried the variant, and therefore the causality of this one was ruled out.”=> There is significant variability in the clinical findings, even between affected members of the same family. Hence, segregation considering only the mother does not seem to be sufficient. I would recommend further segregation at the maternal side of the family. Moreover, I would recommend in silico prediction tools to potentially classify this variant to a class 2 (likely benign), e.g. possible by using http://wintervar.wglab.org/; how about clinvar, gnomAD,…?

Response 5:

Following the recommendation of the reviewer, this suggestion has been changed in the manuscript.

Line 234: However, although some in silico predictors such as SIFT (0.0) and Polyphen-2 (0.999) suggested a possible damaging effect, this variant has been reported as likely benign in ClinVar, and the allele frequency in gnomAD is greater than 0.001. Furthermore, familiar co-segregation studies revealed that the healthy mother carried the variant. Therefore, despite the compatible genotype-phenotype correlation in the patient, this variant was reclassified as likely benign according to ACMG criteria.

Moreover, we have detailed in Material and methods how we have carried out the interpretation of variants:

Line 112: The variants were classified according to the ACMG recommendations and detailed information provided in the public databases gnomAD (https://gnomad.broadinstitute.org/), ClinVar (https://www.ncbi.nlm.nih.gov/clinvar/), dbSNP (https://www.ncbi.nlm.nih.gov/snp/), LOVD (https://www.lovd.nl/), and relevant scientific literature. The in silico analyses were performed using the following online tools: Polyphen-2 (http://genetics.bwh.harvard.edu/pph2/), SIFT (https://sif.bii.a-star.edu.sg/), and the integration tool VarSome (https://varsome.com/).

Point 6:

“As a consequence, the translation is impaired, resulting in a protein of 342amino acids instead of the 377 composing the wild-type protein (Table S2 and Table S3) due to a break in the reading frame caused by the insertion of a premature stop codon.” => very interesting finding!

Response 6:

We truly appreciate the kind remark of the reviewer.

Point 7:

“especially intragenic deletions ranging from single to multiple exons [6,12]. Interestingly, the presence of two to three pairs of microhomology at the breakpoints was found in these cases [12].” => it would be interesting to make a summary/table of all the patients with CdLS and intragenic duplication to see which genes are affected by which duplication and how this affects the phenotype?

Response 7:

Phenotype-genotype correlations are one of the milestones in the research in Cornelia de Lange Syndrome. The incorporation of common criteria in the collection of clinical data (like clinical Score, Kline, A.D et al., Nat. Rev. Genet. 2018, 19, 649-666), use of HPO codes and deep-phenotyping, we are sure it will be very useful to understand these relationships.

Unfortunately, to date it is difficult to unify collected data in the literature. In case of intragenic duplications in CdLS, to our knowledge, only five cases related to CdLS have been reported in the literature. Of these cases, not all of them report complete and detailed information about their clinic. For all these reasons, we believe that the summary/table we can offer at that moment would not meet the agreed quality criteria to scientific community.

And of course, considering the increase of the intragenic duplication cases in CdLS, we have already planned a multi-centre collaborative study. According to common criteria for clinical data collection, we want to generate a comprehensive review of intragenic duplications in CdLS.

Point 8:

Discussion: any limitation(s) of the study? how about the VUS in ANKRD11?... please briefly elaborate.

Response 8:

This case highlights some of the challenges found during the genetic diagnosis processes in CdLS. Throughout the manuscript we have tried to illustrate all the limitations that clinical geneticists encounter during this process: availability of biological samples, technical limitations, variants interpretation, etc. This information has been added in the discussion section of the manuscript:

Line 320: Here we describe the genetic diagnosis procedure of a girl with classic CdLS phenotype, and highlight the limitations faced during this process: (i) Interpretation of VUS, (ii) technical difficulty in detecting small CNVs and somatic mosaic variants, and (iii) functional interpretation of intragenic duplications.

In this study we present a novel intragenic Ì´ 32 Kbp duplication affecting exon 10 of HDAC8 gene. CdLS-associated variants in HDAC8 gene (Xq13.1) are estimated to account for the 4% of the cases of this disorder (CDLS5, OMIM #300882).

Please see the attachment, we upload the edited manuscript.

Reviewer 2 Report

A de novo Ì´32 Kbp intragenic duplication affecting exon 10 of HDAC8  result toCdLS was reported and this found was the contribution of CNVs as a molecular cause underlying CdLS. 

Author Response

Dear Editor,

We sincerely thank to the reviewers for the careful and detailed revision of our manuscript. We have submitted a revised version of the manuscript, addressing the concerns/doubts raised by the referees. Any changes made to the original manuscript have been incorporated using the "Track Changes" function in Microsoft Word. We strongly believe that our manuscript has been improved after the current revision. Please, find below the point-by-point responses to the comments and concerns raised by the reviewers.

We hope these changes will be to your liking.

Best regards,

Prof. Juan Pié, PhD

Response to Reviewer 2 Comments

A de novo Ì´32 Kbp intragenic duplication affecting exon 10 of HDAC8 result to CdLS was reported and this found was the contribution of CNVs as a molecular cause underlying CdLS.

We sincerely thank to the reviewer for the careful and thorough revision of our manuscript. We thank you very much for the suggestion of editing of English language and style required.

We have used the editing services of https://www.mdpi.com/authors/english.

In addition, we have incorporated using the "Track Changes" function in Microsoft Word the comments of Reviewer 1.

We truly appreciate your revision.

Please see the attachment, we upload the edited manuscript.
